# Large-Scale Chromatin Rearrangements in Cancer

**DOI:** 10.3390/cancers14102384

**Published:** 2022-05-12

**Authors:** Kosuke Yamaguchi, Xiaoying Chen, Asami Oji, Ichiro Hiratani, Pierre-Antoine Defossez

**Affiliations:** 1UMR7216 Epigenetics and Cell Fate, Université Paris Cité, CNRS, F-75006 Paris, France; kohsuke.yamaguchi@univ-paris-diderot.fr (K.Y.); xiaoying.chen@parisdescartes.fr (X.C.); 2RIKEN Center for Biosystems Dynamics Research (RIKEN BDR), Kobe 650-0047, Japan; asami.oji@riken.jp (A.O.); ichiro.hiratani@riken.jp (I.H.)

**Keywords:** epigenetics, chromatin, genome organization, transformation

## Abstract

**Simple Summary:**

Cancers have many genetic mutations such as nucleotide changes, deletions, amplifications, and chromosome gains or losses. Some of these genetic alterations directly contribute to the initiation and progression of tumors. In parallel to these genetic changes, cancer cells acquire modifications to their chromatin landscape, i.e., to the marks that are carried by DNA and the histone proteins it is associated with. These “epimutations” have consequences for gene expression and genome stability, and also contribute to tumoral initiation and progression. Some of these chromatin changes are very local, affecting just one or a few genes. In contrast, some chromatin alterations observed in cancer are more widespread and affect a large part of the genome. In this review, we present different types of large-scale chromatin rearrangements in cancer, explain how they may occur, and why they are relevant for cancer diagnosis and treatment.

**Abstract:**

Epigenetic abnormalities are extremely widespread in cancer. Some of them are mere consequences of transformation, but some actively contribute to cancer initiation and progression; they provide powerful new biological markers, as well as new targets for therapies. In this review, we examine the recent literature and focus on one particular aspect of epigenome deregulation: large-scale chromatin changes, causing global changes of DNA methylation or histone modifications. After a brief overview of the one-dimension (1D) and three-dimension (3D) epigenome in healthy cells and of its homeostasis mechanisms, we use selected examples to describe how many different events (mutations, changes in metabolism, and infections) can cause profound changes to the epigenome and fuel cancer. We then present the consequences for therapies and briefly discuss the role of single-cell approaches for the future progress of the field.

## 1. Introduction

Cancer is almost synonymous with genetic changes: tumors accumulate mutations, which enable the emergence of a transformed phenotype [1]. In parallel, transformation is accompanied by widespread changes to the chromatin marks, by which we mean covalent histone and DNA modifications (from here on referred to as “epigenetic changes”). As for mutations, “epimutations” can affect any stage of cancer development, from initiation to progression, to metastasis and resistance [2].

In this review, we will focus on one specific aspect of epigenetic misregulation in cancer: large-scale chromatin rearrangements. In the first section, we will briefly describe the genome and epigenome organization in healthy cells and present some mechanisms that maintain its homeostasy. In the second section, we will illustrate how these mechanisms can go awry and how this contributes to large-scale changes and cancer. In the third (and last) section, we will discuss the therapeutic avenues opened by these mechanisms. We end with a short summary and a discussion of the future evolution of the field.

To maintain focus, we will only very briefly allude to certain topics already reviewed by others in this issue or elsewhere, including phase separation [3] and long non-coding RNAs [4]. We will not attempt to be exhaustive but instead will select examples from the recent literature to illustrate key points.

## 2. The Healthy Epigenome: Its Organization, Dynamics, and Homeostasy

### 2.1. The 3-Dimensional Genome

Our current knowledge of genome organization comes from the convergence of two major approaches: microscopy and molecular biology, with corresponding advances in bioinformatics and modeling [5]. We will provide brief reminders of the notions necessary for the subsequent sections of this review, and we direct interested readers to excellent recent reviews [6,7] for additional details.

At the largest scale, chromosomes occupy defined “territories” in the nucleus (Figure 1). Different chromosomes occupy distinct territories and show a characteristic distance to the nuclear periphery, and this organization is non-random within a cell type. At the 10–100 Megabase scale, the genome is partitioned into two types of compartments: A and B. The A compartments are more euchromatic, internally positioned, interact preferentially with other A compartments, and replicate early, whereas the B compartments are more heterochromatic, associated with the nuclear periphery, interact preferentially between themselves, and are late-replicating. B compartments largely overlap with LADs (Lamina-Associated Domains [8], which are heterochromatic), and vice versa. At a smaller scale, from ~100 kb to a few megabases, one can distinguish Topologically Associating Domains (TADs), i.e., regions with higher probabilities of interaction, separated by interaction boundaries as assayed by Hi-C [9]. Zooming in further still, enhancer–promoter loops are visible in the 10–100 kb range [10].

### 2.2. The One-Dimensional Epigenome and How It Forms the 3D Structure

In parallel with this research on the 3D genome organization, experiments based on approaches such as ChIP-seq, ATAC-seq, and CUT&RUN, have provided us with a wealth of information on the distribution of chromatin features along the one-dimensional genome [11]. The basic unit of chromatin is the nucleosome, formed when DNA wraps around an octamer of histone proteins. The histones can be canonical (H2A, H2B, H3, H4), or non-canonical (also called histone variants, such as H2A.Z or H3.3. Both canonical and non-canonical histones are post-translationally modified. Among the functional landmarks (illustrated in Figure 2) are promoters (marked by H3K4me3), enhancers (H3K4me1/K3K27Ac), transcribed gene bodies (H3K36me3/H3K79me1). Histone variants also regulate genome function [11]. The pattern of histone modifications and histone variants along the genome at any given time results from a dynamic equilibrium between a slew of enzymes that add or remove marks and that form or remodel nucleosomes [11].

DNA methylation is catalyzed by the DNA Methyl-Transferase (DNMT) enzymes DNMT1, DNMT3A, and DNMT3B, and its removal is catalyzed by TET1, TET2, and TET3 [12]. DNA methylation is found mostly on CpG dinucleotides, of which ~80% are typically methylated in healthy cells. The CpGs that escape DNA methylation are in large part found in CpG islands. There are profound mechanistic connections between DNA methylation and histone modifications [13]; two well-known illustrations are that H3K4me3 and CpG methylation is mutually exclusive, whereas H3K9me3 and CpG methylation is frequently associated. In addition, DNA methylation directly influences transcription factor binding [14], which then has consequences on histone modifications.

To summarize, histone modifications, histone variants, and DNA modifications along the chromosomes constitute a regulatory “grammar” that determines genome activity [11]. This picture is continuously being filled in, as new features are being described, such as different types of LOCKs (Large Organized Chromatin Lysine domains, which are often repressed chromatin structures related to cell differentiation) [15] and “Grand Canyons” (Large DNA methylation nadirs) [16]. In addition, the study of chromatin marks is increasingly being carried out at the single-cell level, yielding insight that was unreachable with population-based experiments [17].

A key DNA-binding protein that regulates genome activity on a broad level is the Zinc-finger protein CTCF (CCCTC-binding Factor), an “insulator” protein that prevents activation when placed between an enhancer and a gene in the linear genome [18,19,20]. CTCF, together with cohesin, controls the formation of loops and generates boundaries between neighboring TADs [21,22]. How the 1D genome folds into the 3D genome is being actively investigated [23].

### 2.3. Maintenance Mechanisms: Going through Repairs, Cell Divisions, and Time

The epigenome can be extremely plastic, especially during cellular differentiation. At the same time, it can also be very stable in time: differentiated cells maintain their epigenetic identity throughout their life, they safeguard it even when the underlying DNA is damaged and repaired, and finally they pass it on to their cellular progeny. A set of overlapping mechanisms permit these three feats, and we will discuss them as they are pertinent for the cancer discussion that will follow.

The first aspect, “epigenome repair”, refers to the complex interplay between chromatin and DNA repair [24,25]. The first aspect is that chromatin is a physical obstacle that the DNA repair machinery has to contend with. The second aspect is that local chromatin type influences the choice of DNA repair machinery. For instance, repetitive elements are typically heterochromatic and are preferentially repaired by Non-Homologous End Joining (NHEJ) rather than by Homologous Recombination (HR), which could lead to genome instability. Last, but not least: once DNA repair has been completed, the proper chromatin marks have to be re-established, for instance, repressive marks in heterochromatin. Local chromatin cues are likely to be used to guide the modification of the newly repaired chromatin, but the precise mechanisms remain to be described.

The maintenance of chromatin states through DNA replication and cell division or “epigenome replication” poses a different set of challenges and is the object of very active research efforts [26]. The maintenance of DNA methylation relies on the enzyme DNMT1, which is normally self-inhibited [27,28], until it becomes activated thanks to the action of UHRF1 (Ubiquitin-like with PHD and RING finger domains, 1; Figure 3). UHRF1 itself is a histone modifier, which adds monoubiquitin to histone H3 and other proteins [29], and it interacts directly with the replication machinery [30]. While this basic mechanism is well established, longstanding questions such as the kinetics of DNA remethylation after replication are only starting to be addressed [31,32,33]. Epigenome maintenance is deeply linked to 1D and 3D genome organization: A compartments are more euchromatic, which correlates with early replication timing and a certain set of epigenome maintenance mechanisms which, for instance, seem to rely more on DNMT1/PAF15 interactions [29]. In contrast, B compartments are more heterochromatic, correlating with late replication and different maintenance mechanisms, for instance, UHRF1/H3K9me3 interactions [31] (Figure 3).

Finally, chromatin marks must be robust through time, i.e., through the lifespan of the cell and of the organism. An epigenetic drift occurring with age is well documented; it is easiest to follow on DNA methylation [34], but it likely affects other marks. How the chromatin marks erode with time is a fascinating question that is poorly understood at this point, even though it is very likely to be connected to the previous two points discussed [35].

## 3. Anything That Can Go Wrong, Will Go Wrong: Epigenome Abnormalities in Cancers

In the previous sections, we described actors and processes that establish and maintain the 1D and 3D genome in healthy human cells. This section will illustrate that seemingly any of these mechanisms can go wrong in cancer and contribute to transformation (Illustration in Figure 4). We will take examples from different tumor types. More comprehensive reviews dedicated to epigenetic changes, in particular malignancies, have been published, for instance on pediatric cancers [36] or glioma [37].

### 3.1. Mutation, Overexpression, or Recombination of Epigenetic Writers, Readers, and Erasers

A surprise emanating from cancer sequencing efforts was the realization that epigenetic actors were very frequently mutated, overexpressed, or otherwise activated in tumors, and drove transformation. New cases of deregulated epigenetic actors that drive transformation or sustain the transformed phenotypes are still being discovered [38,39]. For instance, Lung Squamous Cell Carcinomas (LUSC) often display amplification of chromosome 8p11–12, and it was long assumed that the oncogene present in that region was FGFR1, a Fibroblast Growth Factor receptor. However, recent data show that the real culprit is actually a lysine methyltransferase, NSD3 (Nuclear Receptor binding SET domain protein 3). Overactive NSD3 causes an excess of H3K36me2 marks, which rewires the chromatin landscape and promotes oncogenic gene expression [40].

Another particularly interesting recent example is the following. SETDB1 (SET Domain Bifurcated Histone Lysine Methyltransferase 1) catalyzes the formation of H3K9me2 and H3K9me3, and experiments in immortalized MEFs suggested that this activity was necessary to maintain the A/B compartment structure [41,42]. A recent paper investigates in great depth the role of SETDB1 in lung cancer cells [43]. The authors report that SETDB1 expression is required to maintain the transformed phenotype, and that it acts in different ways. In RNA-seq studies, SETDB1 is found to be a transcriptional repressor that removes blocks in a cell migration program, but at a larger scale, SETDB1 has a major influence on the A and B compartments. This, itself, has fascinating consequences on the stiffness on the nucleus, adding to the emerging notion that chromatin has a mechanical effect [44,45].

SETDB1 provides a particularly clear example of a histone methyltransferase having very broad effects in the nucleus of a cancer cells. Many other epigenetic readers, writers, and erasers are affected in cancer [46], and future work will hopefully reveal which ones have equally large-scale effects.

### 3.2. Histone Mutation, Misexpression, or Loss

Histone mutations have also been recurrently identified in cancer sequencing projects, and the role of “oncohistones” is clear in different tumor types, including gliomas, sarcomas, and carcinosarcomas [47]. Mutations in the tail of histone H3, such as H3K27 to Methionine and H3K36 to Methionine, have a global effect on the epigenome by blocking the action of PRC2 (Polycomb Repressive Complex 2) and of SETD2 (SET Domain Containing 2, Histone Lysine Methyltransferase), respectively [47]. More recently, E76K mutations in the globular domain of H2B were described in bladder cancer, as well as in head and neck cancers [48]. Expression of the mutant H2B was sufficient to promote transformation in a model of human breast cancer, possibly because it generates unstable nucleosomes that alter gene expression [48].

In addition to the mutation of canonical histones, the undue expression of specific histone variants also contributes to gene misexpression and transformation. The testis in particular expresses non-canonical histones that permit nuclear condensation during spermatogenesis [49]. Testicular histones are frequently expressed in somatic tumors, and it is speculated that they play a causal role in transformation [50].

While the progress on nucleosomal histones and their variants has been very rapid, the functions of linker histone H1 have been more difficult to elucidate, in part because of the numerous isoforms of the gene. Spearheading work from the Skoultchi laboratory and others revealed that the linker histone was abundant, its isoforms redundant, and that it regulated nucleosome spacing (reviewed in [51]). Loss-of-function mutations in several H1 isoforms have been reported in different types of lymphomas [52]. By developing a mouse model lacking the two H1 isoforms mutated in human lymphoma, Ari Melnick and his colleagues showed that H1 loss caused a global chromatin decompaction in B cells, with many B to A transitions and accompanying changes in histone modifications [53]. This affects, among others, stemness genes that drive the transformation towards lymphoma.

### 3.3. Abnormalities Affecting DNA Methylation Machinery

DNA methylation is conspicuously abnormal in cancer cells, with a low global rate of methylation combined with focal hypermethylation, especially at CpG islands [54]. This pattern has long been recognized and has provided useful markers. For instance, one can examine the DNA methylation pattern of circulating DNA and deduce if an individual might harbor a tumor [55]. Some possible contributors to this pattern have been described: modified histone marks and destabilization of DNMT1 [54]. One aspect that is still relatively poorly understood is the contribution of methyl-sensitive transcription factors and in particular, the few factors attracted by CpG methylation [56]. For instance, the Zinc Finger transcription factors ZBTB4 and ZBTB38 preferentially bind methylated DNA in a sequence-specific manner [57,58]. ZBTB4 is frequently downregulated in cancer [59], and a mouse model shows that a lack of ZBTB4 increases genome instability [60]. The mechanism is yet to be solved and may involve centromeric defects. Low ZBTB38 expression is associated with worse prognosis in prostate cancer [61]; again, the molecular links are not fully elucidated but may involve the function of ZBTB38 in regulating DNA replication [62] or oxidative stress [63].

Of particular interest for this section is UHRF1, a key actor of DNA methylation maintenance. UHRF1 is overexpressed in many cancers, possibly because it is an S-phase marker. Colon cancer is a paradigm in which the roles of DNA methylation, DNMT1, and UHRF1 have been extensively investigated. A minimum threshold of DNMT1 activity is required for colon cancer cells to remain viable [64]. Similarly, UHRF1 is necessary for colon cancer survival, and this is likely to depend on its DNA methylation-promoting function [65]. While UHRF1 is necessary for survival, is its dysfunction sufficient to promote cancer? Very convincing functional experiments show that chronic UHRF1 overexpression in the liver is sufficient to trigger Hepatocellular carcinoma [66,67]. How this occurs is not entirely clear; however, UHRF1 and the related protein UHRF2 seem to destabilize DNMT3A, which may account in part for the low DNA methylation level seen in cancer cells [68]. Another interesting notion is that UHRF1 is involved in the homeostasis mechanisms described earlier, ensuring, for instance, that DNA methylation is properly re-established after DNA damage [69]. Therefore, alterations of UHRF1, affecting its expression, stability, modifications, or partners, may disturb DNA methylation maintenance during replication but also its maintenance outside of S-phase. In addition, the control of UHRF1 activity by cellular metabolites [70] and by alternative splicing [71] is a possible regulatory input that could be miscontrolled in cancer.

In closing, two additional points about UHRF1 are worth mentioning here. The first is that very few mutations of UHRF1 in cancer have been described, suggesting that there may be strong selection for the function of the protein to be maintained. The second is that it is assumed that the main function of UHRF1 is to ensure DNA methylation maintenance [72]. However, another separate function cannot be ruled out. The so-called “histone” modifiers are well-known to have non-histone substrates [73]; similarly, UHRF1 may ubiquitinate and regulate unknown proteins relevant for its function in cancer. In support of this hypothesis, a recent report shows that UHRF1 controls the activity of the kinase AMPK [74]. Further, UHRF1 binds non-histone proteins such as DNA Ligase 1 (LIG1) [75], and it is possible that additional non-histone partners remain to be identified.

### 3.4. CTCF in Cancer

As stated in the earlier sections of this review, CTCF is of paramount importance for gene expression control and chromatin folding. As such, it is understandable that alterations of CTCF are observed in cancer and may provide selective advantages to cancer cells.

First, a number of CTCF binding sites are gained or lost in cancer cells relative to normal cells [76]. This is rarely due to mutations, but rather to chromatin changes (such as gains of DNA methylation blocking CTCF binding) and changes to the transcription factors expressed by the cancer cells. This altered CTCF binding landscape, in turn, modifies the transcriptome and presumably the 3D architecture of the genome.

Second, CTCF has a paralog: CTCFL/BORIS. Interestingly, CTCFL expression is normally restricted to the testis [77]. However, similar to many other germline genes [78,79,80], CTCFL is ectopically expressed in certain tumors, and this affects the function of CTCF [81].

Third, CTCF function is intimately linked with cohesin. STAG2, a cohesin family member, is extremely frequently mutated in cancer. It has recently been demonstrated, in the context of Ewing sarcoma, that STAG2 mutations disturb loop formation and CTCF function [82].

CTCF function is essential for some and maybe all cancer cells. For instance, Myc overexpression drives the formation of super-enhancers that contribute to transformation, and diminishing CTCF activity weakens these super-enhancers and the transformed phenotype [83]. However, CTCF is also essential in normal cells; therefore, the targeting window may be limited.

### 3.5. Viral and Bacterial Perturbations

The study of tumor-inducing viruses has contributed importantly to cancer research [84]. In humans, there is clear evidence that infections with HPV (Human Papilloma Virus), HSV (Herpes Simplex Virus), or EBV (Epstein-Barr Virus), among others, increase the risk of cancer. The effect of these viruses on the cell cycle, apoptosis, or transformation has been extremely well documented and will not be reviewed here. More germane to this review are the epigenetic effects of infections and their role in cancer.

An interesting recent example concerns Epstein–Barr Virus. EBV does not integrate, but instead persists in infected cells as an episome in ~10–20 copies. These copies interact with the cellular genome in a non-random fashion, and “tethering sites” have been described [85], with heterochromatic tethering sites apparently contributing to virus silencing and latent infection. In stomach cancer, it has been found that interaction with EBV episomes can disrupt heterochromatin, activate enhancers, and turn on genes that are pro-transforming [86]. This extends earlier work showing that infection with the bacterium *Helicobacter pylori*, a major facilitator of stomach cancer, also has major epigenetic effects [87]. Recent work on a mouse model with a related *Helicobacter* species (*H. felis*) has shown that the chronic stomach inflammation triggered by the bacteria leads to an increase in DNA methylation by the DNMTs, together with decreased DNA demethylation by the TETs, causing hypermethylation of promoters, including those of tumor-suppressor genes [88].

### 3.6. Metabolism and the Microbiome

Metabolic fluxes have a direct effect on the epigenome. Indeed, the cellular concentration of key molecules used to modify histones and DNA (S-Adenosyl-Methionine, Acetyl-co-A, Ubiquitin, and NAD) is rate-limiting for many of the corresponding enzymes. This concentration is determined by the equilibrium between inputs (diet and generation by the cellular metabolism) and outputs (consumption in various pathways) [89,90]. A spectacular link between metabolism, epigenetics, and cancer is provided by “oncometabolites”, i.e., molecules that accumulate aberrantly because of mutations of metabolic enzymes or because of the metabolic reprogramming following hypoxia [91]. Three such oncometabolites have been described: 2-hydroxyglutarate, succinate, and fumarate; they contribute to a variety of malignancies (glioma, leukemias, neuroendocrine tumors, and renal cancers), and they function by inhibiting αKetoGlutarate–dependent dioxygenases, such as the DNA demethylase TET2 or the histone demethylase KDM4B [92].

Another recent illustration of the links between metabolism, histone modification, and transformation was described in Acute Lymphoblastic Leukemia (ALL). In this context, the mitochondrial activity directly controls the ratio of histone acetylation relative to longer acylations (propionylation, butyrylation, and crotonylation). This has consequences for the distribution of the acetyl-histone binder BRD4 and for cancer gene expression [93].

The contribution of epigenetics to colon cancer has been studied intensely. The rate of colon cancer is clearly linked to lifestyle, and specifically nutrition, which itself influences the gut microbiome. These correlations have been known for some time, but causative links were harder to establish. The transplantation into germ-free mice of human faeces showed that the colon cancer microbiota, in itself, is sufficient to disturb DNA methylation patterns and cause precancerous lesions; this likely occurs via bacterial metabolites [94]. The study of the microbiome in colorectal cancer still yields surprises, such as a potential role of fungi [95]. A long-term hope emerging from these studies is that supplementation of our diets with beneficial metabolites could potentially hinder tumor initiation and/or progression.

## 4. Consequences for Therapy

The enzymes that deposit or remove histone marks are frequently activated in cancer, making them tempting therapeutic targets. To take but one example, the Polycomb Repressive Complex PRC2, responsible for the H3K27me3 mark, is overactive in many malignancies, which has prompted numerous endeavors for the development of clinically useful inhibitors [96]. These treatments have the potential to reverse the large-scale chromatin changes seen in cancer cells or, maybe, to exacerbate them past the point of viability. Tazemetostat, an inhibitor of EZH2, the catalytic component of PRC2, is the first molecule in this class that has gained FDA approval, first for the treatment of epithelioid sarcoma, then for follicular lymphoma [97].

DNA methylation has also long been recognized as an attractive therapeutic target. For over 40 years, the most efficient DNMT inhibitor has been 5-aza-dC (decitabine), successfully used in the clinic to treat Myelodysplasia, Acute Myeloid Leukemia (AML), and Chronic Myeloid Leukemia (CML). This molecule inserts into replicating DNA, where it covalently traps DNMT1. The DNMT1 adduct is then removed by the DNA repair pathway. A new era is opening with the recent description of a competitive DNMT1 inhibitor that does not cause DNA damage, kills leukemic cells in vitro, and outperforms decitabine in mouse models of AML [98].

As discussed above, UHRF1 is an appealing target for drug design, as it contains several druggable protein binding domains, along with a catalytic activity that is essential for the survival of cancer cells. Two groups independently discovered very similar inhibitors targeting the TTD (Tandem Tudor Domain) of UHRF1 [99,100], and another an inhibitor of the PHD (Plant HomeoDomain) finger that is cell-permeable [101]; further chemical and experimental work will be necessary to determine whether these molecules might be useful for cancer treatment.

Inhibitors of DNMTs or of UHRF1 are expected to have global effects on the genomes of both diseased and healthy cells. In contrast, epigenome editing, based on modified CRISPR platforms, is a promising avenue for precise alterations to turn on or off specific genes that have been epigenetically altered [102,103].

Without going into much detail, we will just touch upon phase separation, increasingly recognized as a mechanism that applies to chromatin [104]. It is clearly linked to 1D and 3D genome organization and appears to be involved in the formation of heterochromatin [105] but also in the function of enhancers [106,107]. It is of particular interest for future research that nuclear condensates can actually concentrate therapeutics [108]. Future research might reveal therapeutic agents that affect phase separation.

A closing note on therapies is that, while epigenetic treatments are progressing at great speed, they may still hold their greatest potential when combined with other treatments, including chemotherapy, radiotherapy, and immunotherapy [109].

## 5. Conclusions

Cancer genetics, as developed over the past five decades, have brought us tremendous insight [110]. Key successes include the identification of oncogenes and tumor suppressors, the delineation of transforming pathways operating in different types of tumors, the identification of mutational processes, and the inroads into the “Ecology of cancer”, i.e., the ways tumors evolve and interact with their environment. These conceptual progresses came hand in hand with technical advances in DNA sequencing, bioinformatics, cell isolation, and genome engineering. All of these findings have led to a vast increase of our knowledge but also to concrete improvements into how tumors are detected and treated.

An obvious parallel can be drawn between cancer genetics and cancer epigenetics, a younger discipline that has filled conceptual gaps that could not be explained by genetics alone. It has provided new tools for the early detection of cancer and led to new classes of therapeutic agents, which can be used in combination with, or in addition to, other anti-cancer treatments such as chemo- or immuno-therapy.

Cancer genetics and cancer epigenetics are intricately linked. Genetic events can inactivate or amplify key epigenetic actors such as DNA methyltransferases or histone modifiers. Conversely, the local chromatin structure modulates the rate of mutagenesis [111], and epigenetic silencing is an alternative to mutations for inactivating tumor suppressors. Beyond these local events, chromatin changes underpin large-scale changes in genome organization that drive cancer, as discussed in this review.

There are still challenges ahead for cancer epigenetics. To name just one, distinguishing driver from passenger epimutations is a thorny task. In addition, much of the knowledge that has been gained by the community and is reported in this review has been based on population-based assays. However, innovative single-cell experiments have clearly demonstrated that cancer cells are heterogeneous when it comes to chromatin, and this has functional consequences [112,113].

Single-cell cancer genetics is making great strides. For example, there are ongoing efforts to analyze copy number variations (CNVs) from single-cell RNA-seq data [114,115]. In addition, it has been shown that single-cell genome-wide CNV analysis can be carried out with reasonably good resolution in a haplotype-resolved manner by single-cell DNA replication sequencing (scRepli-seq) [116]. With further increases in resolution and throughput, it should not be difficult to generate a fine-resolution ‘phylogenetic’ map of CNV clonal evolution within tumor tissues, leading to the identification of the key upstream CNV events. Advances of this type, applied to chromatin marks [117], will likely provide important tools to better understand large-scale chromatin rearrangements and their functional roles in cancer.

## Figures and Tables

**Figure 1 cancers-14-02384-f001:**
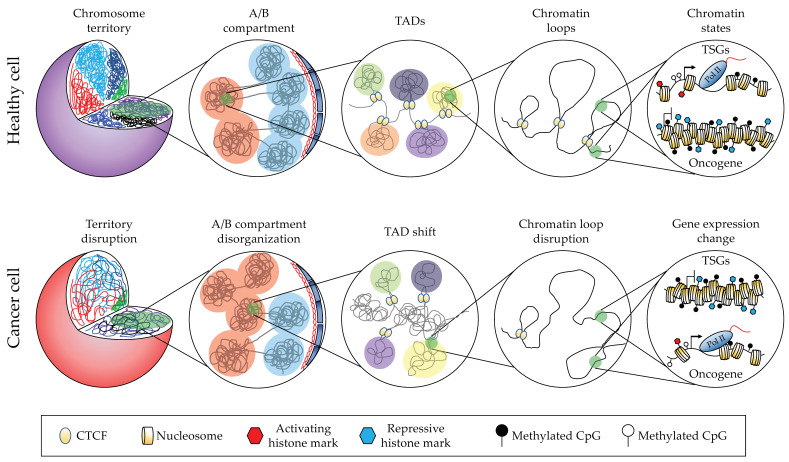
The hierarchical levels of genome organization and how they are modified in cancer cells.

**Figure 2 cancers-14-02384-f002:**
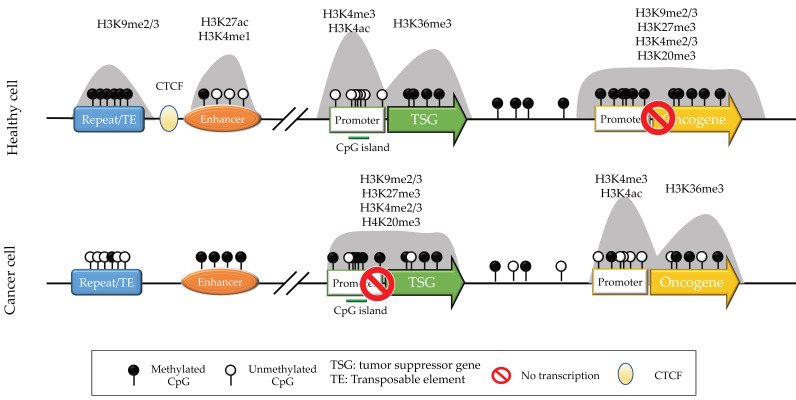
A few selected chromatin marks, their distribution along a portion of the genome, and their alteration in cancer cells.

**Figure 3 cancers-14-02384-f003:**
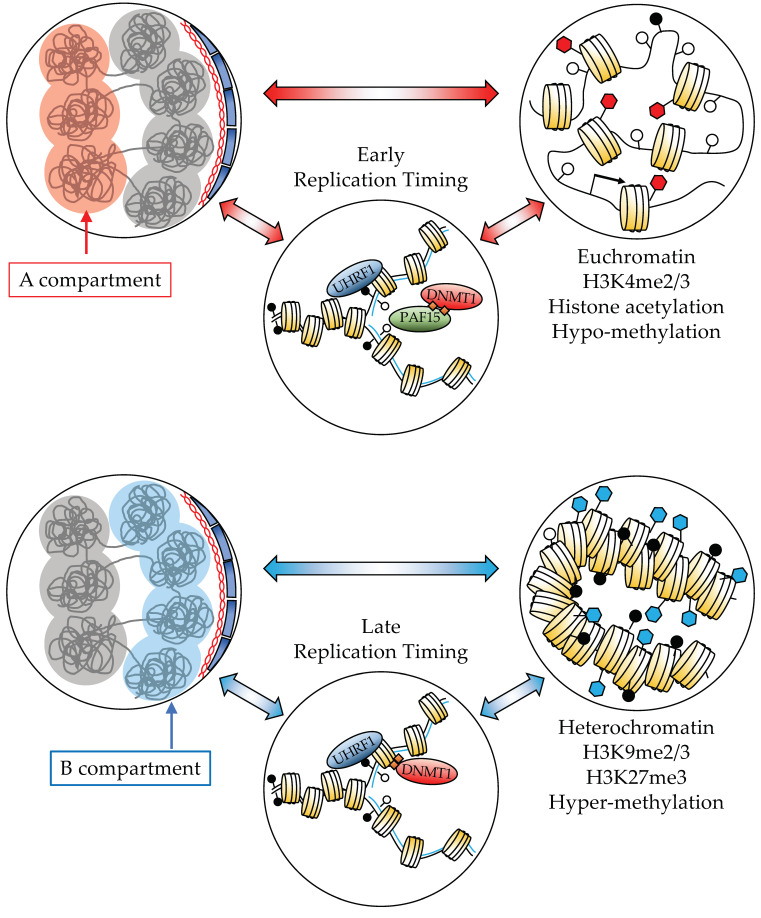
The links between genome compartments, chromatin marks, and epigenome maintenance (histone mark maintenance not depicted, for simplicity).

**Figure 4 cancers-14-02384-f004:**
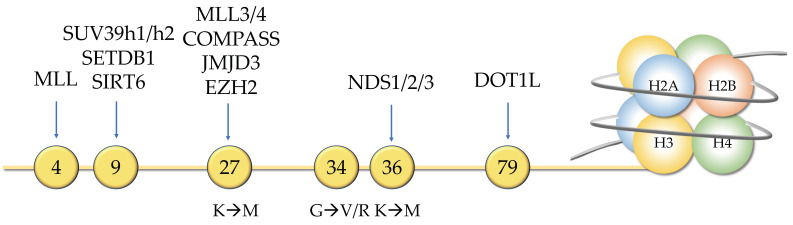
Many writers and erasers of histone H3 are modified in various cancers. In addition, the histone itself can be the target of oncogenic mutations.

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
