# Peer review of "Large-Scale Chromatin Rearrangements in Cancer"

_cancers, 2022, doi:10.3390/cancers14102384_

Round 1

Reviewer 1 Report

Title: Large scale chromatin rearrangements in cancer

Summary: The review summarizes important large-scale chromatin structures, regulatory chromatin modifiers, and more localized histone and DNA modifications that are all necessary for regulating gene expression and maintaining DNA integrity. Following this the authors proceed to describe a variety of ways that changes to this epigenetic homeostasis, including those induced by external agents, can lead to the development of cancers. Tying this together at the end, the authors briefly cover epigenetic therapeutic targets for cancer treatment.

Overall the review is a fairly comprehensive summary of epigenetics structures, regulatory players, how changes in epigenetic homeostasis lead to cancer, and how some of these epigenetic players are therapeutic targets. It is a well designed and thorough review for new and old investigators in the field. Slight edits in the text and figures may help improve the clarity for new investigators in the field, but no major edits are required.

Major comments:

  1. Figure 1 : The figure itself is a nice review of A/B compartmentalization and chromatin loops. While TADs are mentioned, it may be worth highlighting where TADs and LADs fall within the A/B compartmentalization image as they are closer to the center or near the periphery of the nucleus, respectively.

  1. While the authors expand on some mechanisms/actions of certain chromatin structures and regulatory proteins in the second part of the review, some basic definitions may need to be relayed earlier. For example, while A and B compartments are described in relation to their location within the nucleus (line 49-51), their locations are not associated with their chromatin state until much later when they are defined as euchromatic and heterochromatic, respectively (line115-120).

  1. There are several instances of a chromatin structure, histone, or player being mentioned but not given enough context to understand the function it plays in either the first or second part of the review. An example of this is when the authors describe LOCKs. While the acronym is defined, the context may not be obvious for new readers in the field for whom these reviews are critical. Further explaining that LOCKs are often repressed chromatin structures related to cell differentiation may help with reader clarity.

                Minor comments:

  1. Line 52-54: While LADs and TADs are defined, it might be worth mentioning that LADs are heterochromatic while TADs are more open, euchromatic regions.

  1. Bacterial perturbations is listed in the title of section 3.5, but only one example (Heliobacter pylori) is mentioned and is simply said to have major epigenetic effects. It would be good to briefly expand on how this bacterium induces DNA methylation changes (e.g. hypermethylation of tumor suppressor genes).

  1. There are several acronyms that are defined, but several that aren’t. Some examples of this include UHRF1 (line 111), TTD and PHD (line 312), DNMT (line 68), CTCF (line 85), SETDB1 (line 142), PRC2 and SETD2 (line 160-161). While some of these are mentioned only once and may not warrant defining the acronym, some of these are used frequently within the text (e.g. DNMTs, CTCF). Additionally, whereas HR and NHEJ are only mentioned once, both acronyms are defined.

Author Response

Dear Reviewer,

We thank you for your prompt and thorough evaluation of our review. We appreciate the time you have taken.

Your comments are very insightful and we have taken them all into account in the revised version of the review, as detailed in the following point-by-point response.

Best regards,

Pierre-Antoine Defossez, PhD

Summary: The review summarizes important large-scale chromatin structures, regulatory chromatin modifiers, and more localized histone and DNA modifications that are all necessary for regulating gene expression and maintaining DNA integrity. Following this the authors proceed to describe a variety of ways that changes to this epigenetic homeostasis, including those induced by external agents, can lead to the development of cancers. Tying this together at the end, the authors briefly cover epigenetic therapeutic targets for cancer treatment.

Overall the review is a fairly comprehensive summary of epigenetics structures, regulatory players, how changes in epigenetic homeostasis lead to cancer, and how some of these epigenetic players are therapeutic targets. It is a well designed and thorough review for new and old investigators in the field. Slight edits in the text and figures may help improve the clarity for new investigators in the field, but no major edits are required.

 > We thank you for this positive appraisal of the review.

Major comments:

  1. Figure 1 : The figure itself is a nice review of A/B compartmentalization and chromatin loops. While TADs are mentioned, it may be worth highlighting where TADs and LADs fall within the A/B compartmentalization image as they are closer to the center or near the periphery of the nucleus, respectively.

            > Thank you for this suggestion, this has been done.

  1. While the authors expand on some mechanisms/actions of certain chromatin structures and regulatory proteins in the second part of the review, some basic definitions may need to be relayed earlier. For example, while A and B compartments are described in relation to their location within the nucleus (line 49-51), their locations are not associated with their chromatin state until much later when they are defined as euchromatic and heterochromatic, respectively (line115-120).

      > Thank you for this suggestion, we have made the change you suggested. 

  1. There are several instances of a chromatin structure, histone, or player being mentioned but not given enough context to understand the function it plays in either the first or second part of the review. An example of this is when the authors describe LOCKs. While the acronym is defined, the context may not be obvious for new readers in the field for whom these reviews are critical. Further explaining that LOCKs are often repressed chromatin structures related to cell differentiation may help with reader clarity.

            > Thank you for this suggestion, this has been done.

                Minor comments:

  1. Line 52-54: While LADs and TADs are defined, it might be worth mentioning that LADs are heterochromatic while TADs are more open, euchromatic regions.

            > This is now mentioned as you have suggested.

  1. Bacterial perturbations is listed in the title of section 3.5, but only one example (Heliobacter pylori) is mentioned and is simply said to have major epigenetic effects. It would be good to briefly expand on how this bacterium induces DNA methylation changes (e.g. hypermethylation of tumor suppressor genes).

             > We have elaborated on this point, as you suggested.

  1. There are several acronyms that are defined, but several that aren’t. Some examples of this include UHRF1 (line 111), TTD and PHD (line 312), DNMT (line 68), CTCF (line 85), SETDB1 (line 142), PRC2 and SETD2 (line 160-161). While some of these are mentioned only once and may not warrant defining the acronym, some of these are used frequently within the text (e.g. DNMTs, CTCF). Additionally, whereas HR and NHEJ are only mentioned once, both acronyms are defined.

      > Thank you for pointing this out. All the mentioned acronyms have now been      defined.

Reviewer 2 Report

In this review, Yamaguchi, Chen et al., discuss how altered mechanisms of chromatin regulation are linked to cancer. After a brief description of key aspects of genome organization and epigenetic modifications, the authors present examples of epigenome disruption in cancer. Although many aspects of chromatin regulation are not covered (or very succinctly), the authors clearly specify that they are not aiming for a broad and exhaustive review but rather use several examples to illustrate their point. Therefore, I think this is an interesting piece of work whereby the reader can approach how chromatin misregulation may be related to cancer in a fairly straightforward and didactic way. I support the publication of this manuscript, however, I have detailed some minor comments below that I believe would improve it.

Line 24: Using “DNA modifications” when referring to epigenetic changes can be misleading as one might think of DNA sequence modification (i.e., mutations).

Line 34: The authors say that the review ends with a discussion about the application of single-cell approaches but there is no such a discussion.

Line 47: The authors say “The two chromosomes in a pair occupy similar territories, and this organization is stable within a cell type”, which could be misleading since to my knowledge, paternal and maternal copies occupy their own territory rather than to be mixed within a single one.

Figure 1 legend: I suggest to change “Chromosome compartment” to “Chromosome territory” to match common nomenclature and avoid confusion with A/B compartments.

Paragraph 2.2: When introducing histone modifications, it might be instructive to (briefly) link these marks to genome activity. For instance, promoters are not always marked with H3K4me3/ H3K27ac and enhancers with H3K4me1/H3K27ac as suggested in the review (lines 62-63; the presence of these marks suggests that they are active). This would also allow the authors to succinctly present repressive histone marks (H3K27me3 and H3K9me3) since they are not introduced although they are mentioned later in the review (e.g., in Figure 2 and 3).

Figure 2: Stating in the legend that Methylated/Unmethylated refers to DNA methylation (as opposed to histone methylation) will help the reader understand the Figure.

Line 93: The authors say “Cells maintain their epigenetic identity throughout their life, they safeguard it even when the underlying DNA is damaged and repaired, and finally they pass it on to their cellular progeny.” I suggest to rephrase this sentence since the epigenome can undergo substantial modifications, for example during cell differentiation.

Lines 146-148: These sentences are not clear to me.

Line 249: This sentence is not clear to me (are the authors referring to potential therapeutic treatments?).

Author Response

Dear Reviewer,

We thank you for your prompt and thorough evaluation of our review. We appreciate the time you have taken.

Your comments are very insightful and we have taken them all into account in the revised version of the review, as detailed in the following point-by-point response.

Best regards,

Pierre-Antoine Defossez, PhD

In this review, Yamaguchi, Chen et al., discuss how altered mechanisms of chromatin regulation are linked to cancer. After a brief description of key aspects of genome organization and epigenetic modifications, the authors present examples of epigenome disruption in cancer. Although many aspects of chromatin regulation are not covered (or very succinctly), the authors clearly specify that they are not aiming for a broad and exhaustive review but rather use several examples to illustrate their point. Therefore, I think this is an interesting piece of work whereby the reader can approach how chromatin misregulation may be related to cancer in a fairly straightforward and didactic way. I support the publication of this manuscript, however, I have detailed some minor comments below that I believe would improve it.

> We thank you for this positive appraisal of the review.

Line 24: Using “DNA modifications” when referring to epigenetic changes can be misleading as one might think of DNA sequence modification (i.e., mutations).

> Yes, we agree and this has been modified.

Line 34: The authors say that the review ends with a discussion about the application of single-cell approaches but there is no such a discussion.

> Thank you for pointing this out, we have corrected this line.

Line 47: The authors say “The two chromosomes in a pair occupy similar territories, and this organization is stable within a cell type”, which could be misleading since to my knowledge, paternal and maternal copies occupy their own territory rather than to be mixed within a single one.

> We apologize if this was unclear. As you correctly point out, the maternal and paternal chromosomes do not colocalize. We meant to say that they occupy similar positions, ie more internal or more peripheral. This has been rephrased.

Figure 1 legend: I suggest to change “Chromosome compartment” to “Chromosome territory” to match common nomenclature and avoid confusion with A/B compartments.

> Thank you for this suggestion, which we have applied.

Paragraph 2.2: When introducing histone modifications, it might be instructive to (briefly) link these marks to genome activity. For instance, promoters are not always marked with H3K4me3/ H3K27ac and enhancers with H3K4me1/H3K27ac as suggested in the review (lines 62-63; the presence of these marks suggests that they are active). This would also allow the authors to succinctly present repressive histone marks (H3K27me3 and H3K9me3) since they are not introduced although they are mentioned later in the review (e.g., in Figure 2 and 3).

Figure 2: Stating in the legend that Methylated/Unmethylated refers to DNA methylation (as opposed to histone methylation) will help the reader understand the Figure.

> Thank you for this suggestion, which we have applied.

Line 93: The authors say “Cells maintain their epigenetic identity throughout their life, they safeguard it even when the underlying DNA is damaged and repaired, and finally they pass it on to their cellular progeny.” I suggest to rephrase this sentence since the epigenome can undergo substantial modifications, for example during cell differentiation.

> We totally agree and have rephrased the sentence.

Lines 146-148: These sentences are not clear to me.

> Our apologies for the initial lack of clarity. These sentences have been rephrased.

Line 249: This sentence is not clear to me (are the authors referring to potential therapeutic treatments?).

> We imagine the reviewer meant line 349 (and not 249)? If yes, our apologies for the initial lack of clarity. This sentence has also been rephrased.

Reviewer 3 Report

In this review, the authors cover a wide and important field: the role of epigenetics in cancer. Given the vastness of the task, the authors choose some key examples to explain the concepts instead of trying to mention everything that is known. In general, I find the review well written, well structured, and useful to a wide range of readers. However, it can be improved by revising the following:

Mayor points:

  • The introduction should include a definition of the nucleosome as the basic chromatin unit.
  • The abstract mentions that the review will “briefly discuss the role of single-cell approaches for the future progress of the field” (lane 15). While I was excited to read this discussion, in fact it consists of a single sentence at the end of the review (lane 345). It would be desirable to develop this, for example explaining why these new techniques might become important to study this particular topic.

Minor points:

  • Instead of mentioning H2A.Z and H3.3 without relating them to any malignancies, I would define what a histone variant is (lane 64).
  • How are LOCKs and Grand Canyons related to cancer? (lane 79).
  • Figure 1 should include a symbol definition table, such as the one below figure 2. Does the barrel between “Chromatin loops” represent CTCF? In that case, the role of CTCF is different if you choose to focus on the figure or the text: in the text, it is specified that CTCF “generates boundaries between neighboring TADs” (lane 87). This point should be clarified.
  • Figure 3 is not referred to in the text.
  • What is the blue circle contained in the nucleus represented in figure 3? If it is the nucleolus, this should be stated in the figure legend.
  • In the introductory sentence to section 3.1, the authors declare that epigenetic actors drive transformation. The example chosen to illustrate this does not seem to be the ideal, since SETDB1 is “required to maintain the transformed phenotype”, but does not drive a cell from an untransformed state to a transformed one. Nevertheless, I find SETDB1 an interesting example, so I suggest to broad the introductory sentence to include the maintenance of the transformed phenotype. I would also mention another example of a transforming factor.
  • It would be worth mentioning whether there are PRC2 inhibitors and how far are they in the road to clinical use (lane 296).
  • Please define AML and CML (lane 304).

Typos:

  • A full stop is missing both in lanes 151 and 155.
  • Lane 207: “promoter” should be replaced by “promote”.

Author Response

Dear Reviewer,

We thank you for your prompt and thorough evaluation of our review. We appreciate the time you have taken.

Your comments are very insightful and we have taken them all into account in the revised version of the review, as detailed in the following point-by-point response.

Best regards,

Pierre-Antoine Defossez, PhD

In this review, the authors cover a wide and important field: the role of epigenetics in cancer. Given the vastness of the task, the authors choose some key examples to explain the concepts instead of trying to mention everything that is known. In general, I find the review well written, well structured, and useful to a wide range of readers. However, it can be improved by revising the following:

>We thank you for this positive appraisal of our review

Mayor points:

  • The introduction should include a definition of the nucleosome as the basic chromatin unit.

      >This has been modified as suggested

  • The abstract mentions that the review will “briefly discuss the role of single-cell approaches for the future progress of the field” (lane 15). While I was excited to read this discussion, in fact it consists of a single sentence at the end of the review (lane 345). It would be desirable to develop this, for example explaining why these new techniques might become important to study this particular topic.

      >We realize now that we created an expectation that we did not fulfill. We did not want to extend the review too far beyond its current size, so we first lessened the expectation by removing the offending sentence from the introduction. But we agree it is an exciting and timely topic, so we have added some additional text and references on the topic at the very end of the review.

Minor points:

  • Instead of mentioning H2A.Z and H3.3 without relating them to any malignancies, I would define what a histone variant is (lane 64).

      >This has been modified as suggested

  • How are LOCKs and Grand Canyons related to cancer? (lane 79).

      >It is not yet very clear if they are related to cancer. However, in this first part of the review, which deals with the epigenome landscape of normal cells, we thought it was worthwhile to mention these features that not all readers may yet be familiar with.

  • Figure 1 should include a symbol definition table, such as the one below figure 2. Does the barrel between “Chromatin loops” represent CTCF? In that case, the role of CTCF is different if you choose to focus on the figure or the text: in the text, it is specified that CTCF “generates boundaries between neighboring TADs” (lane 87). This point should be clarified.

      >Thank you for pointing this out. We have clarified the symbols. In addition, CTCF is involved in both functions: TAD boundaries and more local chromatin loops. We have made this more clear.

  • Figure 3 is not referred to in the text.

      >This has been corrected, thank you

  • What is the blue circle contained in the nucleus represented in figure 3? If it is the nucleolus, this should be stated in the figure legend.

      >This has been indicated, in the figure legend

  • In the introductory sentence to section 3.1, the authors declare that epigenetic actors drive transformation. The example chosen to illustrate this does not seem to be the ideal, since SETDB1 is “required to maintain the transformed phenotype”, but does not drive a cell from an untransformed state to a transformed one. Nevertheless, I find SETDB1 an interesting example, so I suggest to broad the introductory sentence to include the maintenance of the transformed phenotype. I would also mention another example of a transforming factor.

      >You are right, thank you for pointing this out. We have broadened the introduction, and also mentioned another example of a factor that is transforming (NSD3).

  • It would be worth mentioning whether there are PRC2 inhibitors and how far are they in the road to clinical use (lane 296).

      >A reference has been added to address this question, thanks.

  • Please define AML and CML (lane 304).

      >This has been done, thank you.

Typos:

  • A full stop is missing both in lanes 151 and 155.
  • Lane 207: “promoter” should be replaced by “promote”.

      >All of these typos have been corrected, thank you.
